# Apple Blossom Agricultural Residues as a Sustainable Source of Bioactive Peptides through Microbial Fermentation Bioprocessing

**DOI:** 10.3390/antiox13070837

**Published:** 2024-07-13

**Authors:** Stefano Tonini, Ali Zein Alabiden Tlais, Pasquale Filannino, Raffaella Di Cagno, Marco Gobbetti

**Affiliations:** 1Faculty of Agricultural, Environmental and Food Sciences, Free University of Bolzano-Bozen, 39100 Bolzano, Italy; stefano.tonini2@unibz.it (S.T.); raffaella.dicagno@unibz.it (R.D.C.); marco.gobbetti@unibz.it (M.G.); 2Department of Soil, Plant and Food Science, University of Bari Aldo Moro, 70121 Bari, Italy; pasquale.filannino1@uniba.it; 3International Center on Food Fermentation, 39100 Bolzano, Italy

**Keywords:** fructophilic lactic acid bacteria, yeasts, bioactive peptides, polyphenols, functional activities

## Abstract

This study explored the impact of starter-assisted fermentation on apple blossoms to enhance their potential as a source of antioxidant and antifungal molecules. *Fructobacillus fructosus* PL22 and *Wickerhamomyces anomalus* GY1 were chosen as starters owing to their origin and promising ability to modify plant secondary metabolites. An initial assessment through microbiological and physicochemical analyses showed superior outcomes for starter-assisted fermentation compared to the spontaneous process. Enzymatic hydrolysis of proteins, primarily controlled by starters, orchestrated the generation of new low-molecular-weight peptides. *W. anomalus* GY1 also induced modifications in the phenolic profile, generating a diverse array of bioactive metabolites. These metabolic changes, particularly the release of potentially bioactive peptides, were associated with significant antioxidant activity and marked antifungal efficacy against three common mold species. Our results shed light on the potential of microbial starters to valorize agricultural wastes and convert them into a valuable resource for industry.

## 1. Introduction

In an era where fermentation has acquired a significant role in the generation of new products and in an attempt to exploit the intrinsic value of a wide variety substrates, including agro-food by-products, exploring the potential effects of fermentation on flowers as substrates is intriguing. Apple flowers, abundant in sugars, phenolic compounds, glycoproteins, peptides, and amino acids, have emerged as a valuable reservoir of potential antioxidant components and excellent nutrients for various microorganisms [1,2]. These flowers are often overlooked, despite their diverse functionality. Notably, in apple production, approximately 7% of flowers are necessary for a commercially profitable harvest, with the central flower in the distinctive corymb undergoing maturation into an apple fruit, while the surrounding flowers typically undergo mechanical removal [3]. Considering the global apple harvested area and production, which reached approximately 4.9 million hectares and 86.14 million tons in 2018, substantial quantities of apple orchard waste, including flowers, are generated. The recycling of these residues involves numerous practical requirements and approaches, such as on-site disposal through open combustion, direct mulching, and landfilling, all of which have their inherent ecological risks and social drawbacks [4]. Consequently, apple flowers represent an underutilized by-product that deserves innovative methods to unlock their tapped potential, especially in the food industry and pharmaceutical applications [1,5].

Fermentation is an eco-sustainable biotechnology and time-honored process, in-volving microorganisms that convert raw materials into products with improved shelf life, nutritional, and functional attributes [6]. While the exploitation of apple flowers through fermentation remains unexplored, the extracts derived from various flowers (e.g., rose, calendula, mahua, elderflower, lavender, and hibiscus) are fermented to make wine, tea, and other functional beverages [7,8,9,10,11,12]. Hence, microbial metabolism during the fermentation of apple flowers may lead to the development of novel apple-based fermented products, enriching their bioactive compound profiles and contributing to sustainable food processing.

The influence of microbial activity on the generation of bioactive compounds during fermentation is a critical factor to consider. The deliberate selection of tailored starter cultures offers precise control over fermentation conditions, ensuring the achievement of desired outcomes [13]. In light of recent attention towards bacteria belonging to the genus *Fructobacillus*, which exhibits fructophilic traits and is mainly isolated from flower sources, our focus was directed towards *Fructobacillus fructosus* as a first strategic choice to ferment apple flowers. The metabolic suitability of *Fructobacillus* spp. has been highlighted during the fermentation of cocoa beans, pollens, and sourdough [6,14]. This species has the ability to modify plant secondary metabolites, such as alkaloids and phenolic acids, when it successfully follows different labyrinth paths promoting its growth and adaptive survival [15]. While some studies have reported the antimicrobial and antifungal activities and probiotic effects of *F. fructosus*, insights into its capacity to produce bioactive peptides are limited [16,17]. Given these multifaceted properties, *F. fructosus* was highly suggested as starter culture and/or probiotic in food formulation, with positive effects on the nutritional value of foods, especially those rich in fructose [14,16,18,19].

Among the starter candidates to be used for the fermentation of non-conventional substrates, such as flowers, the metabolic role of yeasts should not be disregarded. Among a wide variety of yeasts, our attention has been directed towards *Wickerhamomyces anomalus*, which possesses noticeable enzymes, such as β-glucosidase and esterase, along with exceptional proteolytic activities [20,21]. Moreover, this yeast species exhibits wide environmental adaptability, with isolates sourced from diverse and harsh niches, demonstrating robust resilience to oxidative, salt, and osmotic stress, and varying pH conditions [22]. Owing to its antifungal metabolites derived from diverse metabolic pathways, *W. anomalus* functions not only as an effective biocontrol agent against gray mold in tomato but also extends the shelf life of wheat flour bread [23,24]. However, despite these promising traits, the application of *W. anomalus* in the valorization of food and by-products remains relatively underexplored [16].

Therefore, we aimed to investigate the potential of *F. fructosus* and *W. anomalus* in advancing the sustainable utilization of apple flower by-products by generating functional ingredients. Employing a pattern of analytical methodologies, we highlighted the metabolic changes in the phenolic compound, amino acid, and potential bioactive peptide compositions during fermentation. Thereafter, we aimed to establish possible associations between these modifications and antioxidant and antifungal activities. In a broad sense, this study sheds light on the promising prospects of fermented apple flower in advancing functional food development and the sustainable utilization of agricultural by-products.

## 2. Materials and Methods

### 2.1. Raw Materials

Apple (*Malus domestica*) flowers of the Red Delicious cultivar were harvested manually from organic apple orchards located in Bolzano, South Tyrol (Italy). The central flower within the characteristic corymb was left to ensure optimal fruiting, while the remaining blossoms were collected. Harvested flowers were then transported to the lab, where they underwent immediate freeze-drying and were stored under refrigerated conditions.

### 2.2. Microorganisms and Culture Conditions

*Fructobacillus fructosus* PL22 isolated from bee-collected pollen [6] and *Wickerhamomyces anomalus* GY1 isolated from *M. domestica* cultivar Golden Delicious (personal data), belonging to the Culture Collection of Micro4Food laboratory of Faculty of Agricultural, Environmental and Food Sciences, Libera Università di Bolzano, Bolzano, Italy, were used as starter cultures. Prior to the fermentation experiments, FLAB and yeast cultures were maintained as stocks in 20% (*v v*^−1^) glycerol at −20 °C and routinely propagated at 30 °C for 24 h in fructose yeast extract peptone (FYP) and Sabouraud dextrose broths, respectively [25].

### 2.3. Fermentation of Apple Flower

The freeze-dried apple flower (AF), after being supplemented with sterile distilled water at a concentration of 90% (*w*/*v*), underwent homogenization using a classic blender (MB 550, Microtron, Milan, Italy). Cells were propagated in their growth medium until the late exponential growth phase (ca. 18–22 h). Then, cells were harvested by centrifugation (10,000× *g*, 10 min at 4 °C), washed twice with 50 mM potassium phosphate buffer (pH 7.0), and singly re-suspended into an AF homogenate to a final cell density corresponding to ca. 7.0 and 5.0 Log CFUs mL^−1^ for FLAB and yeast, respectively. AF fermentation was performed at 30 °C for 24 h, resulting in samples referred to as Fermented-AF. Samples were taken before (Raw-AF) and after fermentation. AF without a microbial inoculum was incubated under the same conditions and used as a control (Unstarted-AF). The microbiological analysis were carried out on different agar media. The monitoring of the starters after fermentation was carried out by RAPD-PCR using the P7 primer for LAB and RP11 for yeast [26].

### 2.4. Carbohydrate, Organic Acid, and Ethanol Quantification

To determine the consumption of carbohydrates and the synthesis of lactic and acetic acids and ethanol, water-soluble extracts (WSE) of Raw-, Unstarted-, and Fermented-AF were used [27]. Briefly, one gram of sample was mixed with 9 mL of Tris-HCl (0.1 mM), pH 8.8, and incubated at 4 °C for 1 h under stirring conditions (200 rpm). After the incubation, the mixture was centrifuged at 12,000 × *g* for 10 min. The resulting WSE was filtered and stored at −20 °C until further use. The concentrations of glucose, fructose, mannitol, lactic acid, acetic acid, and ethanol were determined using high-performance liquid chromatography (HPLC) equipped with an Aminex HPX-87H column (ion exclusion, Biorad, CA, USA), a Perkin Elmer 200a refractive index detector (RI) (IDEX Health & Science, Rohnert Park, CA, USA), and a UV detector operating at 210 nm. Standards were purchased from Sigma-Aldrich (Steinheim, Germany).

### 2.5. Total Protein Quantification

Total protein concentrations of all samples were determined in the water-soluble extracts (WSEs) using the Bradford assay [28].

### 2.6. Low-Molecular-Weight (LMW) Peptide Profiling and Quantification

To separate the active peptide fraction, the WSE was further subjected to ultrafiltration (molecular weight cut-off < 3 kDa) [29]. For this purpose, 15 mL of the sample was loaded in a Vivaspin^®^20 column, 3000 MWCO-PES (Satorius, Florence, Italy), and subjected to centrifugation at 6000 rpm for 80 min. The resulting low-molecular-weight water-soluble peptide extract (LMW-WSE) was used for further investigations.

The peptide concentration was determined by the o-phtaldialdehyde (OPA) assay as originally described by Church et al. [30]. Peptide profiles in LMW-WSE were analyzed by reverse-phase fast-performance liquid chromatography (RP-FPLC) using a Resource RPC column and “AKTA FPLC equipment”, with the UV detector operating at 214 nm (GE Healthcare Bio-Sciences AB, Uppsala, Sweden), as previously described by Pontonio et al. [31].

### 2.7. Identification of Low-Molecular-Weight Peptides by UHPLC/HRMS2

Low-molecular-weight peptides were identified by UHPLC/HR-MS2 (UHPLC Ultimate 3000 with Q Exactive Hybrid Quadrupole–Orbitrap Mass Spectrometer, Thermo Scientific, San Jose, CA, USA) equipped with a C18 column (Acquity UPLC-C18 Reverse-phase, 2.1 × 100 mm, 1.8 µm particle size, Waters Corporation, Milford, MA, USA). The MS data were directly processed with Proteome Discoverer 2.3 (Thermo Fisher Scientific, Dreieich, Germany) coupled with Matrix software (6.0-SP4, Matrix Science, Boston, MA, USA) for peptide sequencing and identification. Peptide and protein identification results were exported after filtering with the Peptide and Protein Validator to achieve a false discovery rate (FDR) below 0.01. Peptides identified in the samples were examined for homology with bioactive peptides identified in the literature using the Bioactive Peptide Database BIOPEP UWM [32].

### 2.8. Amino Acid Profiling

Total and individual free amino acids in the WSE were analyzed by a Biochrom 30 series Amino Acid Analyzer (Biochrom Ltd., Cambridge Science Park, Cambridge, UK) with a Li-cation-exchange column (20 by 0.46 cm inner diameter), as described by Rizzello et al. [33].

### 2.9. Total Free Phenolic Compound Quantification

Total free phenolic compounds were assessed according to the Folin–Ciocalteu method [34]. Analyses were carried out using a methanol/water-soluble extract (MWSE). Two grams of the sample was mixed with 20 mL of a methanol/water solution (70:30, *v*:*v*) and acidified with hydrochloric acid (0.1%, *v*/*v*). The mixture was sonicated (amplitude 60) using a Vibra-Cell sonicator (Sonic and Materials Inc., Danbury, CT, USA) for 1 min (two cycles, 30 s/cycle, 5 min interval between cycles) in an ice bath. Extraction continued for 1 h under stirring conditions at room temperature. Total free phenolic compounds were expressed as milligrams of gallic acid equivalents per gram of fresh weight.

### 2.10. Identification and Quantification of Free Phenolic Compounds

With the objective of elucidating the specific phenolic compounds and investigating the influence of fermentation on the phenolic composition, detailed profiling of free phenolic compounds was conducted within the MWSEs. A targeted LC-MS/MS analysis of 45 free phenolic compounds was performed according to a revised version of the method previously designed and validated by Tlais et al. [35] using an UHPLC Dionex 3000 (Thermo Fisher Scientific, Dreieich, Germany) equipped with a Waters Acquity HSS T3 column (1.8 µm, 100 mm × 2.1 mm) (Milford, MA, USA) and coupled to a TSQ Quantum™ Access MAX Triple Quadrupole Mass Spectrometer (Thermo Fisher Scientific, Dreieich, Germany) with an electrospray source.

### 2.11. In Vitro Antifungal Activity

The hyphal radial growth rate assay was used to determine the in vitro antifungal activity of the MWSE and LMW-WSE [36]. *Penicillium roqueforti* DPPMA1, *Penicillium carneum* CBS112297, and *Penicillium albocoremium* CBS 109582 were selected as mold indicators. A 5 mm Ø of fresh fungal mycelia from the refreshed culture was placed in the center of the mini-plates and used as the inoculum for the antifungal assay. The hyphal radial growth rate was measured after 5 days of incubation at room temperature.

### 2.12. In Vitro Antioxidant Assays

Aiming to determine the radical scavenging capacity, the LMW-WSE and MWSE from all the samples were used for the analysis. DPPH radical scavenging activity was measured using 1,1-diphenyl-2-picrylhydrazyl radical (DPPH^∙^), as previously described by Yu et al. [37].

ABTS radical scavenging activity was estimated by an antioxidant assay kit (Sigma-Aldrich) according to the manufacturer’s instructions. Trolox, a water-soluble vitamin E analog, was used as a control antioxidant [38].

### 2.13. Statistical Analysis

All analyses were carried out in triplicate for each batch of samples. Data were subjected to analysis of variance by the General Linear Model (GLM) of the R statistical package (R, version 4.1.2). Pairwise comparisons of treatment means were achieved by Tukey’s adjusted comparison procedure with a *p* value (*p*) < 0.05.

## 3. Results

### 3.1. Apple Flower Fermentation

The protocol for fermenting AF comprised (i) freeze drying, (ii) re-hydration, and (iii) fermentation at 30 °C for 24 h. Prior to fermentation, the main difference between AF samples was regarding the cell densities of fructophilic lactic acid bacteria (FLAB) and yeasts. As a consequence of the inoculation process, the initial cell densities of FLAB in AF fermented with *F. fructosus* PL22 (PL22-AF) and yeasts in AF fermented with *W. anomalus* GY1 (GY1-AF) were approximately 7 and 5 Log CFUs/g, respectively. After fermentation at 30 °C for 24 h, the cell densities of presumptive FLAB and yeasts in Started-AF increased by 1.76 ± 0.04 and 1.79 ± 0.03 Log CFUs/g, respectively. Presumptive LAB and yeasts were found in Raw-AF (2.72 ± 0.03 and 3.88 ± 0.05, respectively) and Unstarted-AF (5.42 ± 0.05 and 5.22 ± 0.02, respectively) after the incubation. Moreover, after fermentation, RAPD-PCR biotyping confirmed the presence of *F. fructosus* PL22 in PL22-AF and *W. anomalus* GY1 in GY1-AF. Raw-AF had initial pH value of 5.87 ± 0.02, which significantly (*p* < 0.05) decreased to 5.32 ± 0.04 after fermentation with *F. fructosus* PL22 (PL22-AF). After fermentation, the pH value of GY1-AF and Unstarted-AF decreased slightly (5.51 ± 0.02 and 5.61 ± 0.03, respectively).

### 3.2. Analysis of Carbohydrate, Organic Acid, and Ethanol Concentrations

Glucose, fructose, and mannitol were the dominant carbohydrates (13.30 ± 0.04, 13.12 ± 0.03, and 7.67 ± 0.01 mg/g fresh weight (FW), respectively) in Raw-AF (Figure 1a and Appendix A). Throughout fermentation, there was a significant (*p* < 0.05) reduction in glucose and fructose contents across all the samples, albeit to varying extents. After a 24 h incubation, Unstarted-AF had the highest (*p* < 0.05) reduction in glucose levels (58%), followed by GY1-AF (53%) and PL22-AF (35%). Fructose was mainly metabolized by GY1-AF (23%), followed by Unstarted-AF (17%) and PL22-AF (14%). The synthesis of mannitol was only found in AF fermented with *F. fructosus* PL22. The main microbial metabolites were identified as lactic and acetic acids and ethanol. As expected, the synthesis of lactic acid was only found in AF fermented with *F. fructosus* PL22 (1.68 ± 0.01), which also had the highest (*p* < 0.05) concentration of acetic acid (1.45 ± 0.01). Conversely, ethanol was mainly found when AF was fermented with *W. anomalus* GY1 (3.93 ± 0.14) and in Unstarted-AF (2.09 ± 0.18). PL22-AF contained traces of ethanol.

### 3.3. Total Protein and Peptide Profiles

The hydrolysis of proteins and consequent release of peptides were monitored after fermentation using a water-soluble extract (WSE) and low-molecular-weight water-soluble peptide extract (LMW-WSE), respectively. Raw-AF had a protein and peptide concentrations of 5.94 ± 0.33 and 0.31 ± 0.02 mg/g FW, respectively (Figure 1b). After the incubation, both Started- and Unstarted-AF were subjected to hydrolytic activity with a reduction (*p* < 0.05) in protein content (ca. 26%) compared to Raw-AF. The highest (*p* < 0.05) release of peptides was found in PL22-AF (0.73 ± 0.01 mg/g FW), followed by GY1-AF (0.64 ± 0.01) and Unstarted-AF (0.41 ± 0.02). Qualitatively, peptide profiles within the LMW-WSE were assessed utilizing RP-FPLC, coupled with UV detection at 214 nm. Among the samples, chromatographic analyses unveiled a different peptide profile in the intensities of the chromatographic peaks that were particularly evident in samples PL22-AF and GY1-AF (Appendix A).

### 3.4. Peptidomic Analysis

High-resolution mass spectrometry (HRMS) was employed to characterize peptide compositions in LMW-WSE. The full scan mode captures a broad range of mass-to-charge (*m*/*z*) ratios, providing a comprehensive overview of almost all detectable ions within the sample. Based on the detectable ions, peptide sequences were predicted using Proteome Discoverer 2.3 coupled with Matrix software (6.0-SP4) for peptide sequencing and identification. The main parameters used during the identification process were as follows: enzyme, no-enzyme; peptide mass tolerance, ±5 ppm; fragment mass tolerance, ±0.1 Da; and variable modifications, demethylation (NQ), oxidation (M), and phosphorylation (ST). Peptide and protein identification results were exported after filtering with the Peptide and Protein Validator to achieve a false discovery rate (FDR) below 0.01. The NCBI database was employed for identifying and processing MS/MS data. Hence, this untargeted approach provided valuable insights into the impact of fermentation on the generation, nature, distribution, and composition of peptides. The varying proteolytic activities of different starters had substantial effects on both the quantity and diversity of peptides. All peptides identified among raw and fermented AF included sequences ranging from 4 to 43 amino acids, which were related to parental proteins from *Malus domestica*. Overall, these peptides ranged from ca. 400 to 3000 Da, and their distributions along with related information gained from the Label-Free Quantification (LFQ) analysis are reported in Figure 2a. The HRMS analysis, which provides superior accuracy and resolution for peptides smaller than 3 kDa compared to traditional methods, yielded a total of 1797 distinct peptides across all samples, of which only 1176 were present in the Raw-AF. The use of starters resulted in the highest number of peptides, 1583 and 1564 for PL22-AF and GY1-AF, respectively, compared to 1434 for Unstarted-AF. The cumulative abundance of identified peptides in started samples also notably exceeded that of Unstarted-AF (Figure 2b). The fermentation process was relevant to the dynamics of peptide release, contributing to the increased abundance of certain peptides, the hydrolysis of others, and the generation of entirely new peptides that were absent in the Raw-AF. In fact, PL22-AF produced 23 unique peptides and GY1-AF generated 25, with an overlap of 76 peptides between PL22-AF and GY1-AF (Figure 2c). FIVPPPLK was the most abundant peptide found only in PL22-AF (Figure 3a) and FLLGQPA was found in GY1-AF (Figure 3b), while several peptides were shared between the started samples, notably SIPS, SLSP, and TVSP (Figure 3c).

The peptides identified in all samples were cross-referenced for sequence homology against established bioactive peptides (BPs) through the BIOPEP UWM database. Upon querying the database, it was found that none of the 1797 peptide sequences displayed 100% sequence homology with pre-existing, duly recognized BP sequences.

### 3.5. Amino Acid Profile

Before fermentation, the concentration of total free amino acids (FAAs) in Raw-AF was found to be 3670.2 ± 75.3 µg/g FW. Although, fermentation did not cause a significant (*p* > 0.05) variation in the total FAA concentration, the profile of the individual FAAs was changed. The most abundant amino acids in the Raw-AF were Asn and Gln, followed by Phe, Ala, GABA, Arg, and Pro (Table 1). After 24 h of incubation, several amino acids (Ser, Asn, and Gln) were significantly (*p* < 0.05) metabolized in Unstarted-AF and GY1-AF, and to a lesser extent in PL22-AF, whereas Arg was completely depleted. A slight (*p* < 0.05) declining trend was observed for Phe, with only PL22-AF showing comparable (*p* > 0.05) concentrations to Raw-AF. GY1-AF demonstrated significant (*p* < 0.05) reductions in Val and Met, while Unstarted-AF and PL22-AF displayed significant (*p* < 0.05) releases. Remarkably, PL22-AF stood out as the only starter capable of significantly (*p* < 0.05) synthesizing elevated levels of Ala and Ile, while GY1-AF showed an increase in the Glu content. Following fermentation, a consistent upward trend (*p* > 0.05) was observed among all samples for Thr, Leu, and GABA. PL22-AF showed the highest values for Thr and Leu, whereas GY1-AF recorded the highest values for GABA. His was exclusively found in PL22-AF, while Pro showed negligible differences (*p* > 0.05) between Raw-AF and the fermented counterparts.

### 3.6. Total Free Phenolic Content

Compared to that in Raw-AF (31.5 ± 1.72 mg/g FW), the concentration of total free phenolic compounds in the methanol/water-soluble extract (MWSE) decreased significantly (*p* < 0.05) in Unstarted-AF (24.0 ± 2.73). Conversely, started PL22-AF and GY1-AF samples showed lower, but not significant (*p* > 0.05), values (26.7 ± 0.06 and 31.0 ± 0.51, respectively) than Raw-AF.

### 3.7. Phenolic Compound Profile

Free phenolic compound profiles in the MWSEs were analyzed through a liquid chromatography–electrospray ionization–tandem mass spectrometry (LC-ESI-MS/MS) analysis. The highest peaks of 11 phenolic compounds belonging to distinct chemical classes were identified using external standards (Figure 4). Among these, the predominant phenolic compound in Raw-AF was identified as phloridzin (314.9 ± 19.66 µg/g FW), followed by chlorogenic acid (42.22 ± 3.19), hyperoside (30.9 ± 1.18), and phloretin (14.4 ± 0.55). Other identified phenolic compounds ranged from a minimum of 1.55 ± 0.14 (rutin) to a maximum of 10.2 ± 3.59 (ferulic acid). Under our experimental conditions, the interplay of autochthonous microbiota and the inoculum of starter cultures yielded a substantial (*p* < 0.05) variation in the phenolic compound profile compared to that of Raw-AF. Fermentation caused significant reductions in the concentrations of caffeic acid, ferulic acid, and especially phloridzin, which underwent complete metabolic conversion, in contrast to the initial Raw-AF composition (Figure 4). Although displaying an upward trend as compared to that in Raw-AF, only GY1-AF showed significantly higher levels of quercetin (34.16 ± 5.96) and rutin (2.32 ± 0.13). An upward trend was observed for phloretin and kaempferol levels in all fermented samples, with the highest increase (*p* < 0.05) observed in GY1-AF. Other compounds, including chlorogenic acid, epicatechin, and luteolin, did not exhibit a significant (*p* > 0.05) difference among all the samples.

### 3.8. Antifungal Activity

Aiming to assess the potential antifungal activity of LMW-WSEs and MWSEs from Raw-, Unstarted- and Fermented-AF samples, the hyphal radial growth rate assay was carried out using three food spoilage-related indicators: *Aspergillus versicolor*, *Penicillium roqueforti*, and *Penicillium carneum* (Figure 5A). Regardless of the extract type and, albeit with different intensities, the highest (*p* < 0.05) inhibition of the three indicators was found for PL22-AF followed by GY1-AF. For both fermented samples, the highest inhibition was found against *P. roqueforti* and *P. carneum*, especially when the LMW-WSE was assayed. Compared to that in Raw-AF, Unstarted-AF showed (*p* < 0.05) higher inhibition only when the LMW-WSE was used (Figure 5A).

### 3.9. Antioxidant Activity

To evaluate antioxidant capacities of Raw- and Fermented-AF, ABTS (2,20-azino-bis(3-ethylbenzothiazoline-6-sulphonic acid) and DPPH (1,1-diphenyl-2-picrylhydrazyl radical) assays were employed (Figure 5B). Overall, fermentation remarkably (*p* < 0.05) enhanced the antioxidant capacities of AF, especially when *F. fructosus* PL22 was used as starter. The positive effect of fermentation was particularly (*p* < 0.05) visible in the DPPH assay when the LMW-WSE was assayed (Figure 5B). Although the effectiveness of fermentation was confected, the ABTS^+^ scavenging activity was less affected by processing. No or slight variations were found for Unstarted-AF, as determined for the LMW-WSE and MWSE using DPPH or ABTS assays (Figure 5B).

## 4. Discussion

Among the unexplored natural resources, flowers and, in particular, apple flowers emerge as promising substrates to be investigated. To the best of our current knowledge, our study represents a pioneering effort, as it is the first to leverage fermentation to harness the untapped potential of this agricultural residue, transforming it into a reservoir of bioactive compounds [39,40,41]. Despite being labor-intensive and time-consuming, the agricultural technique of flower thinning is commonly used to improve fruit quality and promote consistent bearing over the years. During thinning, around 80% of the apple flowers are mechanically removed and left on the ground as a by-product [3]. This practice is synchronized with the initial flowering stage, a phase characterized by elevated concentrations of primary minerals, amino acids, phenolic compounds, and flavonoids, in contrast to the subsequent full flowering stage [1]. Based on our findings, agar plating revealed the presence of presumptive autochthonous LAB and yeasts within apple flowers, which drive the spontaneous fermentation that occasionally yields undesired characteristics [42]. The use of starter-assisted fermentation is notoriously followed to overcome this limitation, especially if the starter cultures have been isolated from environments that have similar traits to those where they will be inoculated. Our starters are an example of such a case, where *F. fructosus* was isolated from pollen, while *W. anomalus* was isolated from apples.

The initial assessment of the fermented flower validated our concerns on two fronts: the occurrence of spontaneous fermentation and the characteristics of our selected starters. Autochthonous starters were effective at reducing the sugar content in the apple flower. This reduction transpired with a low production of ethanol, the absence of lactic acid, and the appearance of an unknown peak, indicating the presence of a yeast and bacterial mixture, but notably a lack of LAB (Appendix A). Conversely, our starters not only grew well but also showed an exclusionary effect on uncontrolled indigenous microorganisms [43]. This successful adaptation was substantiated by well-documented carbohydrate metabolism pathways typically associated with *F. fructosus* PL22 and *W. anomalus* GY1 [25]. *F. fructosus* employed the 6-phosphogluconate/phosphoketolase pathway (6-PG/PK) for hexose and pentose fermentation, enabling also fructose reduction to mannitol for NAD^+^ co-factor regeneration [44], whereas *W. anomalus* metabolized sugars via glycolysis, producing pyruvate that was further converted to ethanol and CO_2_ under anaerobic conditions [45]. These metabolic pathways resulted in the generation of microbial metabolites, including lactic and acetic acids, mannitol, and ethanol. The effects of these metabolisms on other biochemical processes, such as protein hydrolysis and phenolic compound metabolism, were evident [6].

While both the Started- and Unstarted-AF exhibited significant protein hydrolysis, our starters played a distinctive role in the production of peptides and metabolism of amino acids. The high levels of peptides released by *F. fructosus* PL22 and *W. anomalus* GY1 might be attributed to their unique pools of proteolytic enzymes such as proteases and was affected by the nature of the endogenous apple flowers proteins [46]. This finding was supported by the distinct profiles of FPLC chromatograms and MS data among the fermented samples. Although we highlighted the remarkable biological properties of the fermented samples, our investigation did not reveal any known bioactive peptides from the literature. This is because the screening was conducted using the BIOPEP UWM database, which contains about 4600 bioactive peptide sequences derived mainly from animal and not plant proteins. However, the use of a peculiar protein source is a novel aspect of our research and provides the impetus for further investigations aimed at overcoming the restrictions of the bioactive peptide database in terms of identified and scientifically validated sequences derived from few substrates [32,47,48]. The microbial release of bioactive peptides has been extensively explored in the context of milk and dairy products, but recent strategic efforts have expanded the scope to include a broader array of food sources, with a particular emphasis on plant-based materials [49]. An intriguing finding supporting the presence of bioactive peptides is that, under the conditions of our study, the scavenging ability of Started-AF was more evident towards DPPH∙ than towards ABTS^+^. Since DPPH∙ is soluble in non-aqueous solvents, whereas ABTS^+^ is water-soluble, this discrepancy was likely due to differences in the solubility and dispersive properties of the free radicals and antioxidant peptides [50]. In fact, a typical structural feature of antioxidant peptides is the presence of hydrophobic amino acid residues such as Ala, Val, Leu, Ile, or Trp, especially at their N-terminus [51,52,53]. The low radical scavenging effect towards ABTS^+^ might be related to the high content of hydrophobic aliphatic amino acid residues in protein fragments. Accordingly, the most abundant new peptides found only in PL22-AF and GY1-AF (FIVPPPLK and FLLGQPA, respectively) had Phe at the N-terminus and a grand average of hydropathicity of 0.825 and 0.957, respectively. Similarly, Manzoor et al. [54] reported the exposure of hydrophobic groups in protein fragments after the hydrolysis of apple seed protein, resulting in increased hydrophobicity.

Given the importance of amino acids in counteracting oxidative processes and as antifungal molecules against *Aspergillus* species [55,56], the role of fermentation in modifying the amino acid composition is noteworthy. Typically, microbe cultures utilize amino acids as a source of energy generation, for the maintenance of the cellular redox balance, and for various biosynthetic processes [57,58]. Our starter cultures exhibited a notable peptidase enzymatic capacity, surpassing the native microbiota in their proficiency to effect peptide-to-amino acid conversions. Specifically, compared to Unstarted-AF, PL22-AF showed significantly higher levels of Ile, Leu, His, and Thr, all of which were previously associated with documented antioxidant and antifungal attributes [56]. In the case of GY1-AF, it facilitated the release of Glu, which indirectly contributed to antioxidant defenses through its participation in glutathione synthesis. Operating as a potent antioxidant, glutathione safeguards cells against oxidative damage [59]. It is worth highlighting that, despite comparable potential in unstarted samples, our starter-assisted fermentation significantly enhanced the liberation of additional amino acids. These encompassed hydrophobic aliphatic amino acids (including Ile, Leu, Ala, Val, and Met) known for their DPPH^∙^ scavenging activity, as the DPPH radical interacts effectively with hydrophobic elements [60,61]. Broadening the scope of study to include other bioactivities, the role of GABA as an inhibitory agent within the central nervous system should not be overlooked [46]. Recently, the presence of the *gabD* and *gabT* genes, which encode enzymes involved in the biosynthesis of GABA, has been confirmed within the genome of *W. anomalus* [62].

Microbial-mediated phenolic metabolism is a complex process that may profoundly shape the bioactive content of apple flowers. Throughout fermentation, phenolics undergo various reactions, encompassing degradation, bioconversion, and oxidation [63]. Based on the literature, *F. fructosus* has been revealed capable of utilizing diverse phenolic compounds as electron acceptors, while *W. anomalus* showed promise in utilizing phenolics during apple wine production [64,65]. The change in the phenolic profile observed in fermented samples should not be associated with a decrease in functionality. On the contrary, this change often contributes to the overall bioactivity of the product [66]. The phenolic composition of Raw-AF had minor differences in terms of quantity and composition when compared to the profiles observed for other apple cultivars, with these variations being influenced by factors such as the specific cultivar, environmental conditions, and geographical location [1]. Nevertheless, microbial fermentation played a pivotal role in reshaping the phenolic composition in fermented samples, leading to a distinct pattern of reduction/production according to the microorganisms involved, with *W. anomalus* GY1 demonstrating the most pronounced impact. After the incubation period, the complete breakdown of phloridzin can be attributed to microbial hydrolysis, a process that results in the formation of phloretin and glucose [67]. This hydrolytic mechanism is commonly initiated by enzymes, particularly β-glucosidases, which are prevalent in yeasts and LAB [68]. This likely accounted for the significant increase in phloretin levels in fermented samples, especially GY1-AF [69]. The same enzymes might be involved in the liberation of metabolites, such as rutin, quercetin, and kaempferol, by catalyzing the glycosylation and hydrolysis of their glycosylated precursors [68,70]. However, these flavonoids might be biosynthesized from aromatic amino acids, with phenylalanine serving as the precursor for the flavanone naringenin. Naringenin acts as a crucial scaffold for various other flavonoid subclasses [71,72]. For instance, the synthesis of kaempferol occurs sequentially downstream from naringenin through the catalytic actions of flavanone 3-hydroxylase and flavonol synthase enzymes. Among numerous flavonoids, kaempferol has garnered significant attention due to its profound influence on the regulation of cancer cells [72]. Meanwhile, quercetin and rutin are renowned for their robust antioxidant attributes that are highly effective in mitigating oxidative stress and providing protection against degenerative diseases [73]. In addition, it is worth considering the possible occurrence of further functional derivatives of phenolic acids that were not detected by our target-specific approach. This speculation is based on established knowledge regarding the presence of phenolic acid decarboxylase and reductase activities in LAB and yeasts, which may be consistent with the significant reductions in ferulic acid and caffeic acid levels in the fermented samples compared to the control [39,74].

The biosynthesis of natural compounds, responsible for antifungal and antioxidant activities, holds paramount importance in the fields of both food and pharmaceutical industries [75]. Based on our findings and recognizing the established reputation of apples and their by-products as natural antioxidants, the transformative changes witnessed in the levels of proteins, peptides, amino acids, and phenolic substances during AF fermentation prompted us to explore such functional features [76]. Regardless of the method of extraction of the antifungal components, the two starters exerted a robust antifungal release capacity compared to the controls, demonstrating their efficacy against three typical food-related indicators: *A. versicolor*, *P. carneum*, and *P. roqueforti* [77]. In line with our findings, the literature underscores the role of *F. fructosus* in enhancing the effectiveness of pollen extracts against *P. roqueforti* and other fungal indicators during fermentation [6]. Likewise, *W. anomalus* displays high inhibitory effects on fungal growth and its capability to generate bioactive peptides with potent antifungal properties when harnessed for the fermentation of various protein substrates has been affirmed [24]. Additionally, *W. anomalus* has a significant role in converting phenolic antifungal compounds [23].

In the course of our investigation, establishing a connection between the bioactivity of apple flower and metabolic changes proved to be challenging, due to the diversity of bioactive compound profiles during fermentation. The increased antioxidant and antifungal activities caused by *W. anomalus* GY1 and *F. fructosus* PL22 seemed to be mainly due to the release of potentially bioactive peptides rather than changes in the polyphenolic profile, which occurred significantly only for GY1-AF. However, we cannot rule out the possibility that our targeted approach to phenolic identification may have missed some phenolics that were modified during fermentation and may be associated with the observed bioactivities. Regardless of the perspective taken, the role of fermentation in releasing phenolic compounds, bioactive peptides, and amino acids to enhance antioxidant, antifungal, and other functional properties is undeniable and has been demonstrated with various food wastes, including spent brewers’ grains, tomato seeds, red radicchio leaves, corn germ and bran, and wheat germ and bran [39,78,79]. Hence, the use of our starter remains a promising strategy to further exploit these or similar agro-food by-products, potentially yielding novel metabolites or properties not previously observed with other starter cultures.

## 5. Conclusions

Our study provided a groundbreaking initiative in harnessing starter-assisted fermentation to unlock the latent reservoir of bioactive potential within apple flower by-products, thereby amplifying their antioxidant and antifungal properties. The release of new low-molecular-weight peptides and amino acids, facilitated by both *W. anomalus* GY1 and *F. fructosus* PL22, was suggested as the main contributor to the heightened radical scavenging and antifungal activities observed. *W. anomalus* GY1 also led to a modification in the phenolic profile, promoting the release of phloretin, quercetin, and kaempferol. As a result, these metabolites hold promise for use in pharmaceutical applications and functional foods. Finally, it is important to emphasize that further in-depth research and characterization of plant-based protein sources are still essential, especially when considering bioactive peptides and unique peptides generated in our started samples.

## Figures and Tables

**Figure 1 antioxidants-13-00837-f001:**
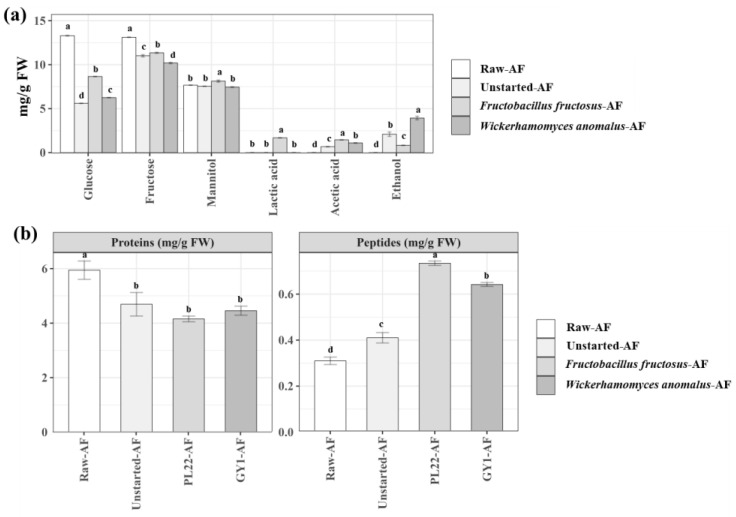
Quantification of carbohydrates, organic acids, ethanol (mg/g FW) (**a**), proteins, and peptides (mg/g FW) (**b**) in raw apple flowers (Raw-AF), AF without the microbial inoculum (Unstarted-AF), and Fermented-AF, which were incubated for 24 h at 30 °C. Fermentations (Fermented-AF) were carried out using *Fructobacillus fructosus* PL22 (PL22-AF) and *Wickerhamomyces anomalus* GY1 (GY1-AF). Data are presented as the means of three analytical replicates ± standard deviations. Bars with different superscript letters differ significantly (*p* < 0.05).

**Figure 2 antioxidants-13-00837-f002:**
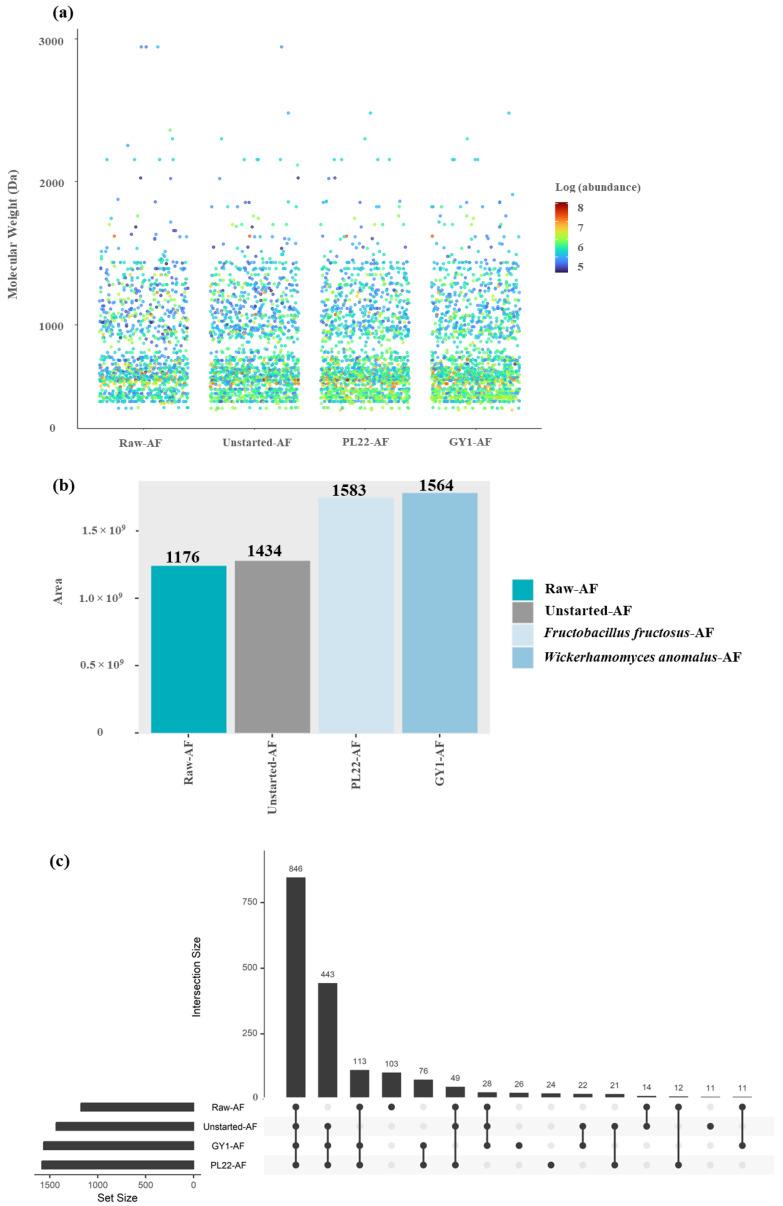
Peptidomic analyses of low-molecular-weight water-soluble extracts (LMW-WSEs) obtained from apple flowers (Raw-AF), AF without inoculum (Unstarted-AF) and Fermented-AF, which were incubated for 24 h at 30 °C. Fermentations (Fermented-AF) were carried out using *Fructobacillus fructosus* PL22 (PL22-AF) and *Wickerhamomyces anomalus* GY1 (GY1-AF). The distribution of identified peptides based on the molecular weight, employing a color scale that transitions from blue to red to represent the Log abundance of each identified peptide within each sample (**a**); total number and abundance of different peptides found in each sample (**b**); and upset plot of the intersection of samples, sorted by identified peptides (dark circles in the matrix indicate sets that are part of the intersection) (**c**).

**Figure 3 antioxidants-13-00837-f003:**
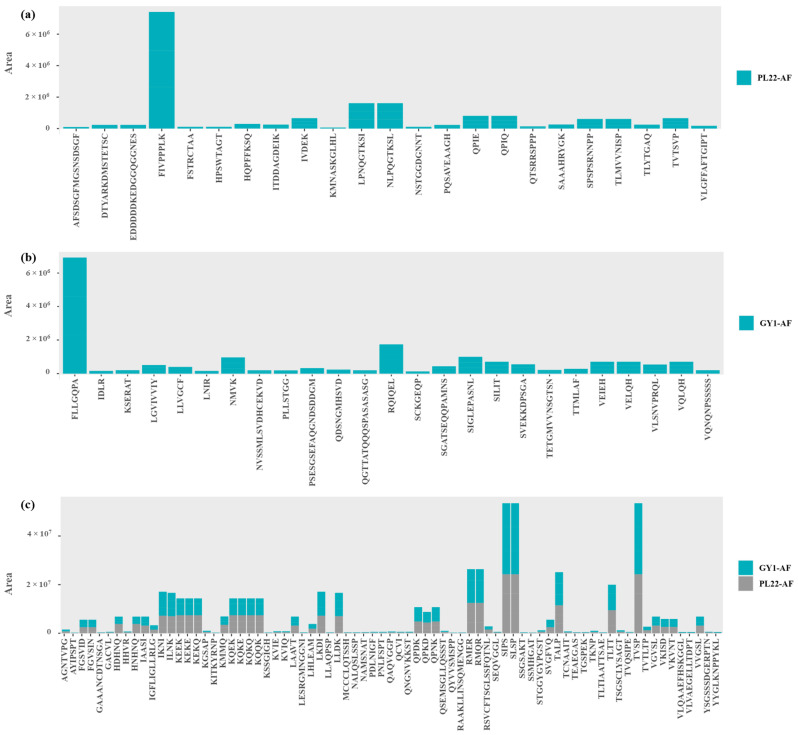
Peptidomic analyses of low-molecular-weight water-soluble extracts (LMW-WSEs) obtained from started apple flowers. Fermentations were carried out for 24 h at 30 °C using *Fructobacillus fructosus* PL22 (PL22-AF) and *Wickerhamomyces anomalus* GY1 (GY1-AF). The relative quantification of unique peptides found only in PL22-AF (**a**); relative quantification of unique peptides found only in GY1-AF (**b**); and relative quantification of unique peptides found only in PL22 and GY1-AF (**c**).

**Figure 4 antioxidants-13-00837-f004:**
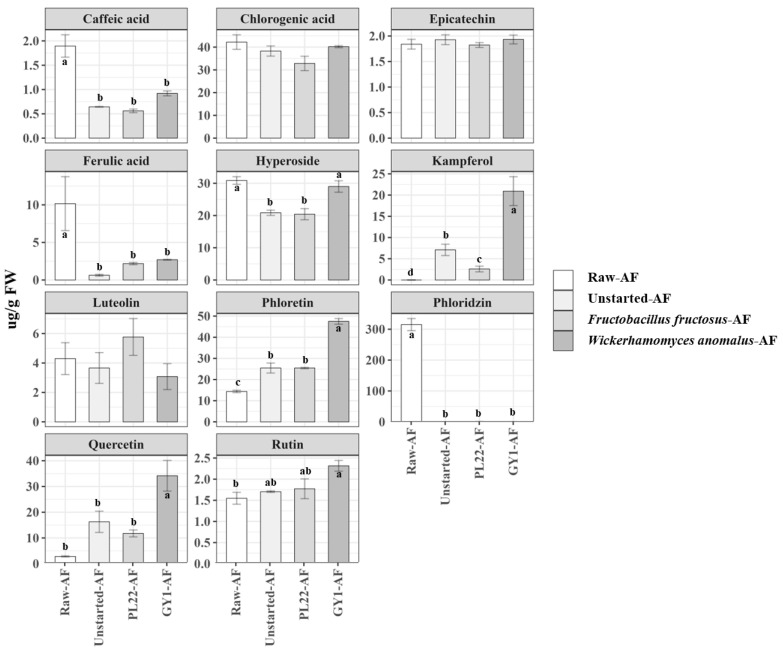
Quantification of free phenolic compounds (µg/g FW) by LC-ESI-MS/MS in the methanol/water soluble extracts (MWSEs) obtained from raw apple flower (Raw-AF), AF without the microbial inoculum (Unstarted-AF), and Fermented-AF, which were incubated for 24 h at 30 °C. Fermentations (Fermented-AF) were carried out using *Fructobacillus fructosus* PL22 (PL22-AF) and *Wickerhamomyces anomalus* GY1 (GY1-AF). Data are presented as the means of three analytical replicates ± standard deviations. Bars with different superscript letters differ significantly (*p* < 0.05).

**Figure 5 antioxidants-13-00837-f005:**
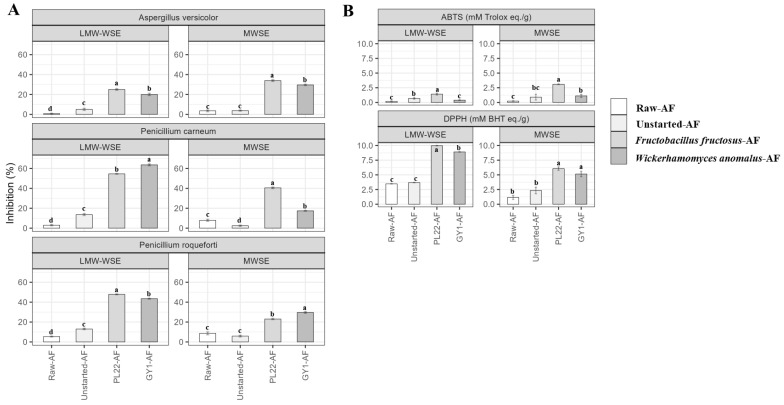
In vitro hyphal radial growth inhibition (%) of *Penicillium roqueforti* DPPMA1, *Aspergillus versicolor* CBS 117,286 and *Penicillium carneum* CBS112297 (**A**), and ABTS (mM Trolox eq./g) and DPPH (mM BHT eq./g) radical scavenging activities (**B**) of low-molecular-weight water-soluble extracts (LMW-WSEs) and MWSEs obtained from raw apple flower (Raw-AF), AF without the microbial inoculum (Unstarted-AF), and Fermented-AF, which were incubated for 24 h at 30 °C. Fermentations (Fermented-AF) were carried out using *Fructobacillus fructosus* PL22 (PL22-AF) and *Wickerhamomyces anomalus* GY1 (GY1-AF). *Penicillium roqueforti* DPPMA1, *Aspergillus versicolor* CBS 117,286 and *Penicillium carneum* CBS112297 without treatments were used as controls. Data are presented as the means of three analytical replicates ± standard deviations. Bars with different superscript letters differ significantly (*p* < 0.05).

**Table 1 antioxidants-13-00837-t001:** Quantification of the total (TFAAs) and individual free amino acids (µg/g FW) in raw apple flower (Raw-AF), AF without the microbial inoculum (Unstarted-AF), and Fermented-AF, which were incubated for 24 h at 30 °C. Fermentations (Fermented-AF) were carried out using *Fructobacillus fructosus* PL22 (PL22-AF) and *Wickerhamomyces anomalus* GY1 (GY1-AF). Data are presented as the means of three analytical replicates ± standard deviations. Rows with different superscript letters differ significantly (*p* < 0.05).

Samples	Raw-AF	Unstarted-AF	PL22-AF	GY1-AF
**Asp**	50.3 ± 0.9 ^b^	84.1 ± 2.4 ^a^	46.0 ± 0.4 ^b^	88.1 ± 0.3 ^a^
**Thr**	32.0 ± 0.5 ^c^	40.8 ± 1.2 ^b^	45.6 ± 0.8 ^a^	42.5 ± 0.3 ^ab^
**Ser**	81.8 ± 1.1 ^a^	0.0 ± 0.0 ^c^	37.9 ± 0.8 ^b^	0.0 ± 0.0 ^c^
**Asn**	1293.9 ± 2.2 ^a^	998.9 ± 13.6 ^c^	1168.6 ± 15.2 ^b^	963.2 ± 2.9 ^c^
**Glu**	6.8 ± 0.7 ^bc^	16.9 ± 3.0 ^b^	0.0 ± 0.0 ^c^	58.4 ± 1.9 ^a^
**Gln**	1167.27 ± 22.25 ^a^	868.62 ± 26.81 ^c^	1011.62 ± 2.85 ^b^	805.7 ± 9.5 ^c^
**Ala**	220.9 ± 12.4 ^b^	270.0 ± 12.1 ^ab^	281.7 ± 4.4 ^a^	231.1 ± 1.8 ^ab^
**Val**	18.9 ± 1.1 ^b^	34.9 ± 0.8 ^a^	40.4 ± 0.8 ^a^	9.0 ± 1.6 ^c^
**Met**	74.0 ± 1.7 ^b^	88.4 ± 2.1 ^a^	90.5 ± 0.9 ^a^	60.1 ± 1.0 ^c^
**Ile**	10.6 ± 1.3 ^b^	16.9 ± 0.5 ^b^	24.1 ± 1.7 ^a^	16.7 ± 0.2 ^b^
**Leu**	0.0 ± 0.0 ^b^	5.0 ± 1.2 ^b^	21.1 ± 2.2 ^a^	3.67 ± 1.5 ^b^
**Phe**	285.5 ± 2.0 ^a^	211.1 ± 8.8 ^b^	284.9 ± 0.8 ^a^	184.2 ± 1.8 ^c^
**GABA**	184.4 ± 4.1 ^b^	786.4 ± 16.6 ^a^	722.6 ± 36.5 ^a^	819.1 ± 3.9 ^a^
**His**	0.0 ± 0.0 ^b^	0.0 ± 0.0 ^b^	4.7 ± 0.78 ^a^	0.0 ± 0.0 ^b^
**Arg**	140.8 ± 1.6 ^a^	0.0 ± 0.0 ^b^	0.0 ± 0.0 ^b^	0.0 ± 0.0 ^b^
**Pro**	103.1 ± 5.3 ^ab^	112.2 ± 4.5 ^ab^	122.3 ± 1.4 ^a^	98.8 ± 2.8 ^b^
**TFAAs**	3670.2 ± 75.3 ^ab^	3534.2 ± 123.9 ^b^	3902.0 ± 7.2 ^a^	3380.48 ± 24.8 ^b^

## Data Availability

The original contributions presented in the study are included in the article/Appendix A, further inquiries can be directed to the corresponding author/s.

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
