# Peer review of "Apple Blossom Agricultural Residues as a Sustainable Source of Bioactive Peptides through Microbial Fermentation Bioprocessing"

_antioxidants, 2024, doi:10.3390/antiox13070837_

Round 1
Reviewer 1 Report
The manuscript of “Apple Blossom agricultural residues as a sustainable source of bioactive peptides through microbial fermentation bioprocessing” is prepared. This manuscript was not well prepared. There are some mistakes. The author should check and revise them. Some comments are indicated as follow.
1.The introduction lacks comprehensive elaboration. It needs to enhance.
2.The unit gram “g” and Centrifugal rate “g” should distinguished.
3.In “P < 0.05”, “P” should be italic.
4.What is “ca. 7 and ca. 5” ? This should be indicated clearly.
5.In Figure 1, the y axis should be started from zero. The other figures should be checked carefully.
6.What do the different letters stand for? This needs to be indicated in different figures.
7.The references should be carefully checked and keep consistent.
The manuscript of “Apple Blossom agricultural residues as a sustainable source of bioactive peptides through microbial fermentation bioprocessing” is prepared. This manuscript was not well prepared. There are some mistakes. The author should check and revise them. Some comments are indicated as follow.
1.The introduction lacks comprehensive elaboration. It needs to enhance.
2.The unit gram “g” and Centrifugal rate “g” should distinguished.
3.In “P < 0.05”, “P” should be italic.
4.What is “ca. 7 and ca. 5” ? This should be indicated clearly.
5.In Figure 1, the y axis should be started from zero. The other figures should be checked carefully.
6.What do the different letters stand for? This needs to be indicated in different figures.
7.The references should be carefully checked and keep consistent.
Author Response
Dear Editor,
I would like to thank you and the referee for giving us the opportunity to improve and resubmit the manuscript entitled "Apple Blossom agricultural residues as a sustainable source of bioactive peptides through microbial fermentation bioprocessing". Please, note that all the recommendations, none excluded, have been considered in the revised version. An itemized list of the revisions according to the referee’s recommendations has been provided.
Kind regards,
Ali Zein Alabiden Tlais
Point-by-point responses to editor and reviewers:
Reviewer 1:
The manuscript of “Apple Blossom agricultural residues as a sustainable source of bioactive peptides through microbial fermentation bioprocessing” is prepared. This manuscript was not well prepared. There are some mistakes. The author should check and revise them. Some comments are indicated as follow.
1.The introduction lacks comprehensive elaboration. It needs to enhance.
Ok, in response to the reviewer's request, additional information and details regarding fermentation, the role of starters, and the contribution of our study to sustainable food systems have been included in the introduction. (P1, L35-36; P2, L45, L49-51, L70-72, L90-92)
2.The unit gram “g” and Centrifugal rate “g” should distinguished.
Ok, the unit gram “g” and the centrifugal rate “g” have been differentiated by italicizing the centrifugal rate “g”. (P3, L114, L128)
3.In “P < 0.05”, “P” should be italic.
Ok, please check the revised version.
4.What is “ca. 7 and ca. 5” ? This should be indicated clearly.
Ok, “ca.” is an abbreviation for the Latin word “circa,” which means “approximately”. This has been better clarified in the manuscript. The inoculum size of the starters (7 and 5 Log CFU/mL) was standardized based on turbidity measurements using a spectrophotometer, which is why we used approximate values to describe the inoculum size.
5.In Figure 1, the y axis should be started from zero. The other figures should be checked carefully.
Ok, we have checked all the figures and confirmed that these plots, generated using R, consistently have the Y-axis starting from zero. Figure 1 and 2 have been modified.
6.What do the different letters stand for? This needs to be indicated in different figures.
Ok, bars in figures and rows in table with different superscript letters differ significantly (P < 0.05). Please check the revised manuscript.
7.The references should be carefully checked and keep consistent.
Ok, all references have been carefully checked.
Reviewer 2 Report
The use of starter-assisted fermentation to valorize agro-food by-products is an innovative approach. It addresses both waste reduction and the creation of valuable new products, which is highly relevant in the context of sustainable food systems. However, it is really scalable, i.e. can the raw materials be collected (effort, amount)?
Specificity: The study focuses on apple flowers, which may limit the generalizability of the findings to other agro-food by-products. Broader studies are needed to confirm if similar results can be obtained with different types of waste materials.
Depth of Mechanistic Insight: While the study reports on the changes in metabolites and bioactive compounds, it may lack in-depth mechanistic insights into how the starters specifically influence these changes. Understanding the underlying biochemical pathways could enhance the robustness of the findings.
A deeper characterization of the flowers would be beneficial. How critical is the time of harvesting them, which effect does storage have?
94 and stored under --> and were stored under
102 Fructose Yeast extract Peptone (FYP) and Sabouraud Dextrose broths--->Can You provide Details or a reference?
196 Trolox-->provide a reference
200 R statistical package.-->provide Version number
Fig. 1. Why is the font size in the legend changing?
519: when considering bioactive peptides and our findings.-->which findings do You mean specifically?
Author Response
Dear Editor,
I would like to thank you and the referee for giving us the opportunity to improve and resubmit the manuscript entitled "Apple Blossom agricultural residues as a sustainable source of bioactive peptides through microbial fermentation bioprocessing". Please, note that all the recommendations, none excluded, have been considered in the revised version. An itemized list of the revisions according to the referee’s recommendations has been provided.
Kind regards,
Ali Zein Alabiden Tlais
Point-by-point responses to editor and reviewers:
Reviewer 2:
Major comments
The use of starter-assisted fermentation to valorize agro-food by-products is an innovative approach. It addresses both waste reduction and the creation of valuable new products, which is highly relevant in the context of sustainable food systems. However, it is really scalable, i.e. can the raw materials be collected (effort, amount)?
Ok, as you rightly mentioned, the valorization of food waste through fermentation is gaining significant interest in sustainable food systems. However, collecting apple flowers, like some other food wastes, poses challenges. The main issue is not the quantity, as we noted in the introduction: only 7% of flowers are needed to generate apples, leaving a substantial amount as waste. An orchard can produce a high volume of flowers depending on its size and the number of apple trees. In south Tyrol, the area under apple trees is approximately 18,400 hectares, so this by-product is substantial. The more problematic aspect is that apple flowers do not naturally fall off the tree in large quantities before fruit sets, necessitating manual collection to improve fruit quality and promote consistent bearing over the years. This process is labor-intensive, time-consuming, and requires careful handling to avoid damaging the flowers. Despite these challenges, optimizing the collection process is feasible to reduce the labor and effort involved. (P11, L338)
Specificity: The study focuses on apple flowers, which may limit the generalizability of the findings to other agro-food by-products. Broader studies are needed to confirm if similar results can be obtained with different types of waste materials.
Ok, in response to the reviewer's request, the discussion now includes broader studies on other agro-food by-products (such as brewer’s spent grains, tomato seeds, red chicory leaves, maize germ and bran, and wheat germ and bran), emphasizing the role of fermentation in releasing phenolic compounds, bioactive peptides, and amino acids, thus maximizing the antioxidant, antifungal effects, and other functional properties. (P15, L530-536)
References
- Mechmeche, F. Kachouri, H. Ksontini and M. Hamdi, Production of bioactive peptides from tomato seed isolate by Lactobacillus plantarum fermentation and enhancement of antioxidant activity, Food Biotechnol, 2017, 31, 94-113.
- Verni, E. Pontonio, A. Krona, S. Jacob, D. Pinto, F. Rinaldi and C. G. Rizzello, Bioprocessing of brewers’ spent grain enhances its antioxidant activity: Characterization of phenolic compounds and bioactive peptides, Front Microbiol, 2020, 11, 1831.
- Z. A. Tlais, G. M. Fiorino, A. Polo, P. Filannino, and R. Di Cagno, High-value compounds in fruit, vegetable and cereal byproducts: an overview of potential sustainable reuse and exploitation, Molecules, 2020, 25, 2987.
Depth of Mechanistic Insight: While the study reports on the changes in metabolites and bioactive compounds, it may lack in-depth mechanistic insights into how the starters specifically influence these changes. Understanding the underlying biochemical pathways could enhance the robustness of the findings.
Ok, further mechanistic insights into the influence of starter cultures on metabolite and bioactive compound alterations and mechanisms behind such changes were thoroughly clarified in the discussion section. (P12, L410-417, L421, L454-455; P13, L468-469, 493-497)
A deeper characterization of the flowers would be beneficial. How critical is the time of harvesting them, which effect does storage have?
Ok, the timing of harvesting is crucial as it must be carried out during a specific period of the season and on certain days. However, as previously mentioned, the collection process could be optimized using more practical techniques, especially for large orchards. In our study, the storage was not an obstacle because the flowers were freeze-dried immediately after collection, preventing any possible oxidation and then were stored under refrigerated conditions.
Detail comments
94 and stored under --> and were stored under
Ok, the sentence has been modified. (P3, L99)
102 Fructose Yeast extract Peptone (FYP) and Sabouraud Dextrose broths--->Can You provide Details or a reference?
Ok, a reference has been added. (P3, L108)
196 Trolox-->provide a reference
Ok, a reference has been added. (P4, L197)
200 R statistical package.-->provide Version number
Ok, the version number has been added. (P4, L201)
Fig. 1. Why is the font size in the legend changing?
Ok, the plots generated using R cannot italicize the names of the starters in the legend. Consequently, when we edit them in PowerPoint, this slight discrepancy arises.
519: when considering bioactive peptides and our findings.-->which findings do You mean specifically?
Ok, the sentence has been revised. Our findings were referring mainly to the unique peptides generated in the started samples. (P14, L549)
Round 2
Reviewer 1 Report
The manuscript "Apple Blossom agricultural residues as a sustainable source of 2 bioactive peptides through microbial fermentation biopro- 3 cessing" has made some modifications. However, some problems were not corrected.
1.In Figure 1, the y axis should be started from zero. The other figures should be checked carefully. This was not revised according to the requirement.
2.The references should be carefully checked and keep consistent. For example,31,32, 76, 77, 78...
The manuscript "Apple Blossom agricultural residues as a sustainable source of 2 bioactive peptides through microbial fermentation biopro- 3 cessing" has made some modifications. However, some problems were not corrected.
1.In Figure 1, the y axis should be started from zero. The other figures should be checked carefully. This was not revised according to the requirement.
2.The references should be carefully checked and keep consistent. For example,31,32, 76, 77, 78...
Author Response
Dear Editor,
Thank you once again for the opportunity to improve and resubmit our manuscript entitled "Apple Blossom agricultural residues as a sustainable source of bioactive peptides through microbial fermentation bioprocessing". We have addressed and checked all the remaining issues, none excluded. An itemized list detailing the revisions made in accordance with the referee's recommendations has been provided.
Kind regards,
Ali Zein Alabiden Tlais
Point-by-point responses reviewers:
The manuscript "Apple Blossom agricultural residues as a sustainable source of 2 bioactive peptides through microbial fermentation biopro- 3 cessing" has made some modifications. However, some problems were not corrected.
1.In Figure 1, the y axis should be started from zero. The other figures should be checked carefully. This was not revised according to the requirement.
Ok, we would like to highlight again that all the plots in the manuscript have their y-axes starting from zero. However, the script used in R to create these plots caused a small gap between the x-axis and the zero level. Nevertheless, we made efforts to eliminate this space and have carefully reviewed all the figures according to the requested requirements. All the figures have been modified.
2.The references should be carefully checked and keep consistent. For example,31,32, 76, 77, 78...
Ok, the listed references, along with others, were checked again and modified when required.